# The Bioeconomy and Food Systems Transformation

Eduardo Trigo [1], Hugo Chavarria [1], Carl Pray [2], Stuart J. Smyth [3,*], Agustin Torroba [1], Justus Wesseler [4], David Zilberman [5] and Juan Martinez [1]

1 Inter-American Institute for Cooperation on Agriculture (IICA), Ipis 2200, Costa Rica
2 School of Environmental and Biological Sciences, Rutgers University, New Brunswick, NJ 08901-8554, USA
3 Department of Agricultural and Resource Economics, University of Saskatchewan, Saskatoon, SK S7N 5A2, Canada
4 Agricultural Economics and Rural Policy Group, Wageningen University, Droevendaalsesteeg 4, 6708 PB Wageningen, The Netherlands
5 Department of Agricultural and Resource Economics, University of California, Berkeley, CA 94720-3310, USA
* Correspondence: stuart.smyth@usask.ca

**Abstract:** While the global number of people experiencing food insecurity remains stubbornly high, innovations have been increasingly adopted that are contributing to ensure that food systems are as resilient and flexible as they can possibly be. Bioeconomy and biotechnology innovations have contributed to improving rural development and food production. Genomic knowledge is an important part of innovative bioeconomy and biotechnology research as it is applied to increase the efficiency of crops, animals, biofuel, bioplastics and bioenergy production. This allows food systems to transform to be more sustainable and equitable, providing healthy, nutritious food, while creating livelihood opportunities and reducing negative impacts. This article highlights the beneficial impacts of innovative bioeconomy and biotechnology products in technologies, particularly as they relate to the Americas.

**Keywords:** biotechnology; genetic modification; innovation; Latin America; research and development; supply chain

## 1. Bioeconomy Concepts and Contributions

The most widely recognized definition of bioeconomy was proposed in the Global Bioeconomy Summit 2018 framework: "bioeconomy is the production, utilization and conservation of biological resources, including related knowledge, science, technology and innovation, to provide information, products, processes and services across all economic sectors aiming toward a sustainable economy" [1]. Bioeconomy policy frameworks and development approaches make use of materials and energy found in biodiversity, biomass, and genetic resources. The knowledge generated about biological principles and processes can be replicated in new product designs.

The bioeconomy concept as a development approach is driven by advances in science and technology (S&T) and the need to address new problems and concerns. Recently, this approach has been advanced by progress in research and development in biological sciences and by complementarity and convergence with the S&T of materials (especially nanotechnology) and information (e.g., artificial intelligence (AI), digitalization, information and communication technologies (ICT), Internet of Things (IoT)). The bioeconomy concept has been favored by concerns associated with climate change, since material replacement and energy-based production processes are essential components of actions needed for adaptation and mitigation, and is seen as an important complement to the decarbonization of the economy. Interest in the bioeconomy concept as a development approach also emerges from societies' concern regarding the ability to meet the increased demand for more sustainably produced food.

In addition, there are increasing changes toward sustainable consumer lifestyles, where consumers are better informed and inclined to buy environmentally friendly products. These changes create opportunities for the utilization of biomass (agricultural residuals, food waste) to increase recycling and to shorten supply chains, but also as an alternative feedstock for the production of numerous materials from fuels and energy to chemicals, bioplastics and pharmaceuticals, among others. Future bioeconomy innovations are expected to generate greater positive impacts on sustainability, such as synthetic biology, novel nitrogen-fixing crops, nanofertilizers, and more.

The bioeconomy concept as a development approach has similarities to and differences with concepts of the circular and green economies, which are included as approaches to sustainable development [2,3]. All are multidimensional concepts, having as goals: the reduction of greenhouse gas (GHG) emissions, energy and material use efficiency, responsible consumption, the importance of social inclusion and the relevance of innovation. However, the bioeconomy is distinguishable by its focus on innovation and transformation of production structures, because its material and energy base are biological resources, including the use of knowledge for processing and the creation of value-added chains (Figure 1).

The bioeconomy makes important contributions to sustainable economic growth from environmental and social points of view, especially in rural areas. For example, the European Union (EU) bioeconomy (post-Brexit composition) employed ~17.5 million people, generating €614 billion of value-added production in 2017 [4]. Also, in 2017, Latin American countries such as Argentina generated 2.47 million direct bioeconomy jobs [5]. Nordic countries have experienced bioeconomy-related employment growth of 5–15% [6]. It is estimated that this development model has an economic potential of US$7.7 trillion by 2030 [7]. Previous projections are supported by trends in the bioeconomy markets. While commodities such as vegetable oil, sugar and cereals have growth rates of less than 4.45%, sectors with higher value added, such as biofuels, bioplastics, and biofertilizers grew by 25%, 20% and 14%, respectively [8]. Using new S&T to add value to biological resources leads to more profitable and sustainable markets. Cingiz et al. [9] demonstrate the various linkages between the bioeconomy sectors, highlighting those that contribute 30% and 50% to the total value added of EU bioeconomy.

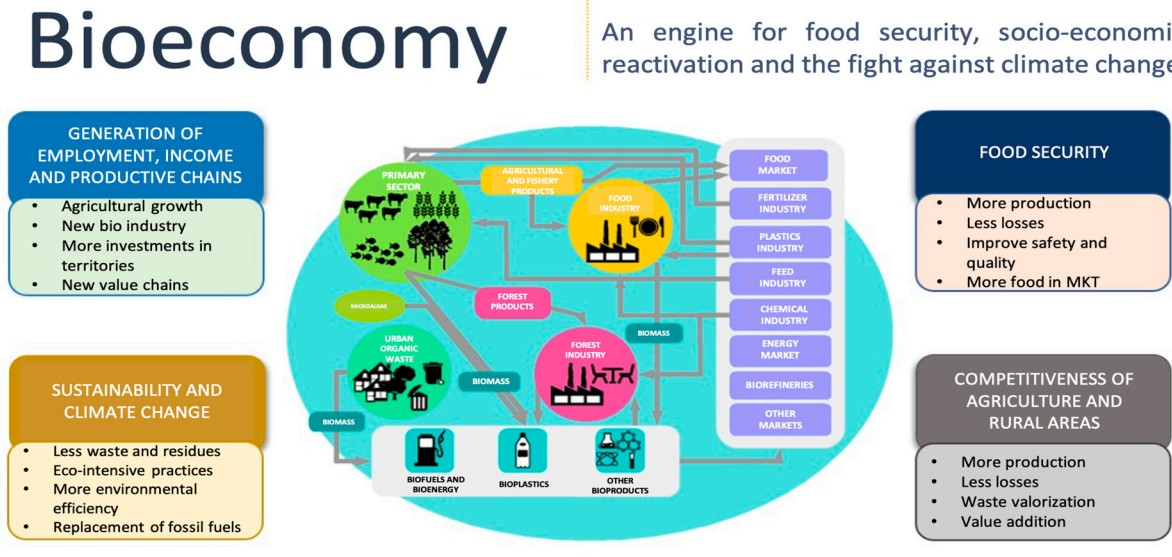

**Figure 1.** Sectors and networks of the bioeconomy. Source: Adapted from [10].

Finally, links between the bioeconomy and the 2030 Agenda for Sustainable Development are demonstrated by using the Sustainable Development Goals (SDGs) as indicators for bioeconomy monitoring and evaluation [11]. In an analysis of national bioeconomy

strategies [12], topics related to the SDGs were indirectly related to objectives, planned actions and proposed measurements for policy instruments aimed at promoting the bioeconomy. Fourteen SDGs relevant to the bioeconomy were identified. The bio-based economy can play a fundamental role in the decarbonization of the planet (SDG 13: Climate Action) and production of agricultural bio-inputs, healthy food and sustainable intensification of agricultural production (SDG 2: Zero Hunger, SDG 3: Good Health and Well-being and SDG 15: Life on Land). Additionally, the closure of production cycles through residual-biomass use improves the sustainable production indicators (SDG 12: Responsible Consumption and Production and SDG 11: Sustainable Cities and Communities). Another contribution of this new paradigm is the design of biomaterials and production of different types of bioenergy (SDG 9: Industry, Innovation and Infrastructure and SDG 7: Affordable and Clean Energy), which help generate new jobs (SDG 8: Decent Work and Economic Growth)

The bioeconomy approach as a development model, which allows achieving the SDGs related to food security and nutrition; health and well-being; and clean water and sanitation, among others, is analyzed in Table 1.

**Table 1.** Potential contributions of the bioeconomy to the SDGs.

| Potential Contribution | SDGs That Contribute |
|---|---|
| Productive models that take advantage of science and technology to use biological resources sustainably and efficiently to make substitutes for petrochemicals (for example, bioenergy, biofertilizers or bioplastics) or to satisfy new consumer demands (for example, functional foods of biocosmetics). | SDG 2: Sustainable Food Production SDG 3: Good Health and Well-Being SDG 7: Affordable and Clean Energy SDG 9: Industry and Innovation SDG 13: Climate Action |
| Use of productive practices that contribute to environmental sustainability and resilience while adding productivity and efficiency. | SDG 13: Climate Action SDG 15: Sustainable Use of Land Biodiversity |
| Circular economy production systems, through the productive use of waste biomass derived from production and consumption processes. | SDG 11: Sustainable Cities and Communities SDG 12: Responsible Consumption and Production |
| Development of products, processes and systems replication processes and systems overserved in nature. | SDG 9: Industry and Innovation SDG 14: Sustainable Use of Underwater Biodiversity SDG 15: Sustainable Use of Land Biodiversity |
| Bioremediation to face environmental contamination problems (for example, recovery of degraded or contaminated soils and treatment of water for human consumption and wastewater. | SDG 6: Clean Water and Sanitation SDG 15: Sustainable Use of Land Biodiversity |
| Increase in the economic density of rural territories from new industrialization processes and local use of biomass for the generation of bioproducts and bioservices. | SDG 8: New Sources of Decent Work and Sustainable Economic Growth |

Source: [13].

## 2. Bioeconomy Contributions to Food Systems Transformation

The transformation toward more sustainable and equitable food systems seeks to provide healthy, nutritious food, while creating livelihood opportunities and reducing negative impacts. To achieve this goal, the United Nations (UN) Food Systems Summit has established five Action Tracks relating to the bioeconomy. **Action Track 1** seeks to ensure the availability of safe, nutritious food for everyone. This requires increasing crop and livestock yields through sustainable intensification activities in multifunctional landscapes, the diversification of production, and good soil management. **Action Track 2** is the shift to healthy and sustainable consumption patterns. In this case, the bioeconomy

can strengthen local value chains, promoting the reuse and recycling of food resources. **Action Track 3** aims to optimize natural resources in food production, processing and distribution as pollution, soil degradation and loss of biodiversity are reduced. For this, the bioeconomy strategies focus on value chains with integrated cycles, which increase efficiency and recycling through products and co-products in different biological systems. **Action Track 4** includes strategies for integrating chains and adding value to products at the local level, contributing to poverty reduction by creating new rural jobs. **Action Track 5** promotes resilience in the face of vulnerabilities, impacts and stresses in food systems. Resilience can be strengthened by a growing bioeconomy, based on the diversification of agricultural commodity production; increased use of bio-based inputs in agriculture; and the diversification of rural incomes into rural production of bioenergy, bio-based industry and environmental services. The current contingencies caused by COVID-19 and recent natural disasters highlight the importance of innovations to prepare food systems for future pressures.

### 2.1. Advantages of Disruptive Scientific and Technological Developments

Advances in biology, ICT and engineering are repositioning the role played by biological resources and improving our ability to understand and take full advantage of the opportunities offered. In recent decades, biology advances have accelerated with new research tools such as CRISPR-Cas9, building on new knowledge of plant, animal and microbial genomes as well as big data [14,15]. Knowledge increases are used to increase the efficiency of crops, animals, biofuel, bioplastics and bioenergy production. They highlight the full potential of the intrinsic value of natural and biological processes. The impact of these transformative trends is augmented by the interaction among them, what is beginning to be referred to as 'technological convergence' [16]. By interacting with each other, different disciplines—biology, biotechnology, chemistry, nanotechnology, data science, ICT, engineering, etc.—are driving the progress of each specific field, blurring the traditional boundaries between economic sectors and changing the competitive advantages of countries and their businesses [17–19].

Information and communication technologies and digitalization are important determiners of economic organization and competitiveness. Widespread connectivity, satellite technologies, data science and artificial intelligence mechanisms, robotics, autonomous systems, electronic and biological sensors, virtual and augmented reality, the IoT and blockchain apps are increasing the efficiency of agriculture, food and biomass supply chains, reducing waste and resource use while increasing the quality of food and biomass [20]. It is also becoming possible to predict climate phenomena and generate risk management programs to better deal with the consequences and monitor climate impacts, which can reduce farm management costs.

Through the use of S&T, the bioeconomy makes it possible to improve productivity and sustainable use of biological resources by developing more productive, disease-resistant and environmentally friendly varieties of plants and animals [21,22]. Science and technology increases biomass productivity and develops new bioproducts with high value-added, such as nutraceuticals, bioenergy and other biological materials used by the cosmetic, pharmaceutical, chemical and other industries. Furthermore, it attaches greater value to biodiversity and generates a range of new services, such as integrated pest management based on biological pesticides and fertilizers [23,24]. It contributes to increasing the efficiency of converting biological resource for food, feed and other uses by improving biorefinery processes.

Technological convergence is a trend contributing to the renewed, modernized vision of agriculture and food systems, value-added chains and international trade, particularly because of young people's technological skills—which exceed those of previous generations—and the need to halt the migration of young people from rural territories to urbanized areas. These new technological scenarios are already beginning to be reflected in

agriculture, agribusiness and the rural milieu, and are increasingly perceived as offering the basis for the development of 'sustainable intensification'.

Technological convergence supports SGDs: 3 (Good Health and Well-being); 8 (New Sources of Decent Work and Sustainable of Economic Growth); 9 (Industry and Innovation) 11 (Sustainable Cities and Communities); 12 (Responsible Consumption and Production); and 15 (Sustainable Use of Land Biodiversity).

### 2.2. Transforming Rural Environments, Generating Income and Employment Opportunities

One key bioeconomy issue is the implications of moving from fossil to bio-based value chains. Fossil raw materials are relatively homogenous and are extracted in high volumes from selected productive deposits of a limited area. They are transformed into products for energy sector materials, the multi-stage chemical sector and the construction sector through large-scale industrial and logistical infrastructures [25]. In contrast, biological carbon—biomass—comes from a highly decentralized context due to the diverse nature of agriculture and forestry and 'does not travel well'. Due to its large volumes, limited shelf-life and low energy and carbon density, it is not economical to transport biomass long distances before processing. Integrated biomass processing facilities need to be organized in a decentralized way, close to raw material sources.

It is these bio-based value chain characteristics that allow for significant transformations of rural landscapes and how they integrate into the economy. Bio-based value chains bring new activities into rural landscapes, diversifying income sources and the nature of existing employment opportunities. Greater economic density generates opportunities for Latin American and the Caribbean (LAC) territories that are highly impacted by situations of unemployment, informality (76% of those employed), poverty (45%, several times more than urban rates) and exclusion. The use of biomass for new industries increases economic opportunities for both agricultural and non-agricultural sectors (which in LAC generate 58% of the income of rural territories) [26].

Outmigration to urban centers, aging populations and lack of youth interest to remain in farming vis-a-vis the promise of a more 'attractive' future in non-agricultural jobs is a common concern in rural communities around the world. According to a 2018 Organisation for Economic Cooperation and Development (OECD) study that included 24 developing countries, only 45% of rural youth are satisfied with their employment. Among the reasons for seeking a new job, rural youth mentioned: a better income (36.7%); greater stability in contracts (20%); better working conditions (17%); and an opportunity to increase skills (13%) [27].

A second strategic component of the bioeconomy concept as a development approach and its impacts on transforming rural environments is the implications of improved energy availability to attract other economic activities beyond bio-based value chain activities. Previously, rural electrification stimulated local development processes and bioenergy options could lower costs through the decentralization of costly energy grids, improving environmental performance through more integral use of residual biomass and waste [28,29]. This is important for regions such as LAC, where forest biomass is equivalent to half of its land area (and 25% of the worlds' forests). Furthermore, it is a region where more than 120 million tons of food are wasted annually (55% of fruits and vegetables, 40% of roots and tubers, 25% of cereals, meats and dairy products, etc.) [30]. Cingiz et al. [9] show the linkages with up- and downstream sectors making up between 30% to 50% of the added value of the bioeconomy in the EU.

An affordable, stable energy supply is a critical restriction to economic development and the bioeconomy is increasingly offering it through options that are not competitive with food production [31,32]. In an increasingly interconnected world, emerging bioeconomy networks are viable strategies for reversing rural outmigration. In 2018, bioenergy generated 3.18 million jobs—equivalent to 30% of all jobs in the renewable energy sector. Moreover, the employment generated by the biofuels sector worldwide is highly concen-

trated; LAC accounts for 50% of liquid biofuel jobs worldwide, while North America accounts for 16% [33].

Improving rural economies through bioeconomy and bio-based energy contributes to supporting SDGs: 3 (Good Health and Well-being); 7 (Affordable and Clean Energy); 8 ((New Sources of Decent Work and Sustainable of Economic Growth); 9 (Industry and Innovation); 11 (Sustainable Cities and Communities); and 15 (Sustainable Use of Land Biodiversity).

### 2.3. Improving Food Chain Resource Use

The diversification in biomass use to produce biofuels contributes to GHG reduction, generates added value and employment and contributes to a safer, more efficient agri-food systems. Biomass fractionating results in a series of biomaterials of different added values. Biomaterials are liquid, solid and gaseous biofuels, which under the term 'bioenergy' represent 10% of the world's primary energy supply [34]. A wide range of products linked to animal and human food (flour protein, expeller, bagasse, distillers dried/wet grains, etc.) and other high value-added products linked to the pharmaceutical, alcohol chemical and oleo chemical industries are also produced.

Biomass fractionation leads to an industry categorized as 'multi-product', in which the production of co-products facilitates a better distribution in raw material production costs, making the system more efficient [35]. Safer agri-food systems are generated, as biofuels serve as a buffer of raw materials that can be use as food in case of crisis or crop losses. The production of biofuels has generated more stable demands for raw materials, generating additional sales channels. According to Torroba [33], 16% of corn production worldwide, 20% of sugar production, 19% of soybean oil and 16% of palm oil are destined toward biofuels. When the prices of related commodities are not attractive, the redirection of raw material derived from crops can be particularly beneficial to farmers. It generates more stable demand for raw materials, creating positive impacts on prices, benefiting neglected LAC groups such as family farmers, of whom 60 million work in the sector.

Biofuel productivity has improved, reflecting learning-by-doing and ongoing technological updating. Processing costs of US corn ethanol declined by 45% between 1983–2010, while production volumes increased seventeen-fold; learning-by-doing and economies of scale played important roles in reducing these costs. Similarly, the cost of producing sugarcane ethanol in Brazil declined by 70% between 1975–2010 [36]. With advances in biotechnology to enhance the productivity of feedstock plants, the efficiency of refining and the use of residue, the cost of biofuels and their environmental impacts will decline, while their added value is enhanced [37].

Enhancing the utilization of resources in supply chains supports SGDs: 7 (Affordable and Clean Energy); 9 (Industry and Innovation); and 13 (Climate Action).

### 2.4. Improved Nutrition and Health

Growing consumer interest in products with natural ingredients promotes new value chains associated with tropical biodiversity. Agro-forestry systems with native fruit trees and traditional forest foods can provide the necessary macro- and micro-nutrients needed to improve nutrition and food security. Crops have increased yields, contributing to higher household incomes, poverty reduction and enhancement of household food security [38]. Biofortified genetically modified (GM) crops have been improving the nutritional quality of food, including increasing proteins (canola, corn, potato, rice, wheat); improving oils and fatty acids (canola, corn, rice, soy); increasing vitamin contents (potato, rice, strawberry, tomato); and increasing mineral availability (lettuce, rice, soy, corn, wheat) [39]. Nutritionally enhanced foods are preventing and/or treating leading causes of death such as cancer, diabetes, cardiovascular disease and hypertension.

In many instances, improving macro-nutrients (proteins, carbohydrates, lipids, fiber) and micro-nutrients (vitamins, minerals, functional metabolites) has significant childhood health improvements, such as reducing blindness due to the lack of vitamin availability. Improved food nutrient content, especially the increase in mineral availability, contributes

to improved immunity systems and reduces stunting [40]. In many developing countries, plant-based nutrient intake accounts for 100% of an individual's nutrient diet, further highlighting the importance of nutritionally enhanced crop-derived foods. Health benefits are extended to adulthood through reductions in cancer causing mycotoxins, such as is found in GM corn.

One quality of life health improvement that has resulted from the small land-holder adoption of GM crops is the reduction in drudgery [41]. The majority of weed control in developing countries is done by hand labor. Hand weeding is labor commonly assigned to women. Gouse et al. found hand weeding was reduced by three weeks over the course of a year with GM corn adoption. This allowed women to have larger vegetable gardens.

Improved biofortification of food and health benefits from biotechnology support SDGs: 1 (End Poverty); 2 (Sustainable Food Production); and 15 (Sustainable Use of Land Biodiversity).

### 2.5. Improved Environmental Sustainability and Climate Resilience

Bioeconomy and biotechnology investments have made substantial environmental improvements, offering the potential to be a leading strategy in efforts to mitigate climate change. It is estimated that biomass could save 1.3 billion tonnes of $CO_2$ equivalent emissions per year by providing 3000 terawatt-hours of electricity by 2050 [42]. It is necessary to establish national instruments of measurement for GHG emissions throughout the life cycle of biofuels according to the different raw materials used to corroborate the environmental advantages. Bio-based products release fewer GHGs compared to fossil carbon commodities.

Another sustainable bioeconomy contribution is the reduction and use of food waste. In the agro-industrial sector in LAC, food waste is around 127 million tonnes/year, enough to satisfy the nutritional needs of 300 million people [43]. Thanks to S&T advances, multiple technologies allow the reduction of waste and its use to produce new bioproducts for the food, energy, chemical, pharmaceutical and construction industries. Food waste can be considered as a cheap feedstock for producing value-added products such as biofertilizers, biofuels, biomethane, biogas and value-added chemicals [44]. These new industries have the potential to contribute to the mitigation objectives of climate change and the environmental sustainability of productive commercial activities due to the substitution of products of fossil origin with a high carbon footprint.

The commercialization of herbicide tolerant canola, corn and soy in the mid to late 1990s revolutionized land management practices, resulting in tens of millions of acres transitioning to zero tillage. The additional commercialization of insect-resistant corn, cotton and soy has resulted in millions of fewer pesticide applications. The reduction in tillage and chemical applications has produced a significant environmental benefit, with 24.6 billion kg fewer carbon dioxide ($CO_2$) emissions and 749 million kg fewer chemical active ingredients being applied [45,46]. Reduced input costs and higher yields have contributed to GM crops providing economic benefits of US$261 billion over the 25-year period from 1996 to 2020 [47]. It is estimated that insect-resistant crops reduced global pesticide use by 37% [48]. Not only are there fewer GHGs emitted during the production of crops, the continuous cropping of fields with no tillage is increasing the soil's sequestration and storage of $CO_2$. Research following 25 years of GM canola production in Saskatchewan found that the combination of effective weed control provided by glyphosate in conjunction with GM canola is driving improved agricultural sustainability [49]. Conventional agricultural practices that require the use of tillage for weed control are estimated to have a net global warming potential that is 26–31% higher than zero tillage land [50]. The adoption of GM technology in corn, soybean and cotton reduced agricultural land and input use, saving 0.15 Gt of GHG emissions, equivalent to roughly one-eighth of the emissions from automobiles in the US [51].

The reduction or elimination of disturbance to the soil layers in a minimum tillage or no tillage system also economically benefits farmers by reducing soil erosion, which

has substantial effects on agronomic performance. Bakker et al. [52] estimates that in mechanized agriculture, for every 0.1 m of soil loss, crop yields are reduced by 4% in the EU and North America. No tillage systems leave the majority of crop residues on the soil surface instead of incorporating them into the soil profile, which helps to increase soil organic matter content, decreasing the negative impacts of erosion. Additionally, crop residues on the soil surface will reflect sunlight and conserve moisture by lowering the temperature of the soil and protecting it from high evaporation levels [53,54].

One emerging and vital area of innovative bioeconomy research is the use of innovative breeding technologies, including gene editing, to improve the abilities of plants to sequester increased amounts of $CO_2$, allowing agricultural food production to make significant contributions to reducing the impacts of changing climates. Changes in a plant's ability to photosynthesize can have additional yield-enhancing benefits. Bioeconomy photosynthesis research that results in plants sequestering greater volumes of $CO_2$ and higher yields will ensure that crop production levels do not decline in the face of changing climates. Transgenic research targeting increased soybean photosynthesis, which contributes to mitigating climate change, has resulted in yield increases of up to 33% [55].

Plant breeding involving biotechnology and gene editing is also providing additional sustainability benefits by developing new varieties that are resistant to diseases that are threatening to destroy species. Fungal diseases and viruses have had devastating impacts on the production of coffee, where an estimated 60% of all production is threatened [56]. Similar circumstances exist regarding the production of bananas, oranges and cocoa. The technology is also being applied to reintroduce species into regions where they were previously made extinct due to disease, such as the case of the American chestnut tree.

The application of biotechnologies that improve environmental sustainability and climate resilience supports SDGs: 2 (Sustainable Food Production); and 3 (Good Health and Well-being).

### 2.6. Upscaling Biotechnology Innovations

Humanity is facing major challenges, including climate change, food security and rural development. The bioeconomy is poised to play a central role in addressing these challenges. New technologies in life and information sciences, combined with practical knowledge of production practices and ecosystems, can unleash the bioeconomy's potential. This requires significant investment in basic and applied research, training highly skilled professionals and a fluid relationship between academia and industry. Zilberman et al. [57] suggest that the 'educational industrial complex' has been essential in establishing the biotechnology and information technology sectors in the USA and throughout the world. In the educational industrial complex, publicly supported basic research within universities and other research institutions leads to discoveries and innovations that are transferred to, and expanded by, startups and other private-sector actors. Their development efforts lead to products that are produced and marketed by the private sector and transferred to final users. The educational industrial complex has already led to the establishment of supply chains for new products, including biofuels and oils, fine chemicals, pharmaceuticals and foods. University researchers have led some of these new ventures, and the exchange between universities and the private sector in clusters such as the San Francisco Bay Area, St. Louis, Davis, Sao Paolo, San Diego, Austin, Mendoza, Santiago, etc.

The supply chains that emerge from these industrial clusters provide direct employment in the production of technological devices and even greater opportunities in the industries resulting from these technologies. The resulting bioeconomy industries are more likely to be concentrated in rural regions, alleviating rural poverty. For example, biofuel and fine chemical production can transfer rents from owners of non-renewable resources such as fossil fuels to the expanded agri-food sector. Biorefineries operate at lower temperatures allowing for constructions smaller in size in comparison to refineries converting fossil fuels. This allows for more diversified as well as spatially distributed scaling-up [58].

The success of the educational industrial complex depends on maintaining academic and research excellence. The pioneering knowledge produced by EMBRAPA was key to the emergence of Brazil as an agricultural powerhouse, suggesting that support for outstanding research institutes linked with industry is a sound social investment.

The three main obstacles to the development of the biofuels sector are regulatory uncertainty, high transaction costs and financial constraints. Upscaling and applying new knowledge requires a science-based regulatory environment that aims to reduce regulatory burdens and accelerate the development and application of new, safe technologies. The emergence of entrepreneurial startups is more likely when venture investors and capital markets are established to support new industries and when regulatory procedures are streamlined to reduce the cost and time needed to establish the venture.

Greater efficiencies in the commercialization and adoption of biotechnology innovative techniques and products contributes to SDGs: 7 (Affordable and Clean Energy); 9 (Industry and Innovation); and 15 (Sustainable Use of Land Biodiversity).

### 3. Moving Forward

Food systems, the "activities involved in producing, processing, transporting and consuming food" [59], are an integral part of the bioeconomy concept as a development approach. New developments in the biological sciences allow countries to address the many challenges that society is facing. We have summarized the many opportunities that the biological sciences have to offer. The translation of these opportunities into practice will not be trivial. There are a number of institutional factors that delay or even prevent the full exploitation of the opportunities the bioeconomy has on offer.

First, the development of research capacity at universities and government institutes can turn these opportunities into technical and social innovations. Second, industries that generate employment and economic growth should be developed based on these innovations and the supply chains,. Third, regulations of innovations should be developed that protect society but do not disrupt the application of these opportunities in production, transportation and consumption and unnecessarily restrict sustainable growth, jobs and resilience. The differences in regulations in different countries often reflect different societal norms and values. These institutional barriers are difficult to solve by one country alone. The UN Food Systems Summit brought together many countries and people to discuss the removal of institutional barriers.

Further to the second point above, bioeconomy developmental and expansion will require private sector participation. Firms of all sizes have contributed to the successful commercialization of GM crops, not to mention the production of value-added products utilizing GM crops as raw ingredients. Public sector innovation plays an important incentive role in the research and development phases of the innovation pipeline, but the private sector is fundamental for successful product or technology commercialization.

Our overview has shown that a great deal can be achieved by building research capacity and reducing institutional barriers. The impacts will be beyond the food systems and affect other sectors of our economies as well. An open discussion will be needed that takes differences in norms and values into account without discriminating against one another. The UN Food Systems Summit provided the opportunity. The results depend on us.

**Author Contributions:** All authors contributed equally to this review and have read and agreed to the published version of the manuscript.

**Funding:** This research received no external funding.

**Institutional Review Board Statement:** Not applicable.

**Informed Consent Statement:** Not applicable.

**Data Availability Statement:** Not applicable.

**Conflicts of Interest:** The authors declare no conflict of interest.

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
