# Peer review of "The Bioeconomy and Food Systems Transformation"

_sustainability, doi:10.3390/su15076101_

Round 1

Reviewer 1 Report

Some suggestions to improve the paper..Table 1 should be retyped to fit the norms of table presentation (closed in the borders and properly typed).

Item 2.5 may include a more recent assessment of GMO impacts., once the primary goal of the paper is to provide a comprehensive view of bioeconomy.

Maybe, the paper could include some remarks about the role of corporations in the bioeconomy development.

Author Response

Some suggestions to improve the paper. Table 1 should be retyped to fit the norms of table presentation (closed in the borders and properly typed).

The table has been recreated and reformatted.

Item 2.5 may include a more recent assessment of GMO impacts., once the primary goal of the paper is to provide a comprehensive view of bioeconomy.

This section has been revised with increased discussion of GM impacts and the resulting benefits.

Maybe, the paper could include some remarks about the role of corporations in the bioeconomy development.

A new paragraph has been added to the concluding section that discusses the importance of the role played by the private sector.

Reviewer 2 Report

Dear Authors,

The presented review characterizes the global world problems quite deeply. The material of the manuscript is built logically. I think the manuscript will be of interest to a wide range of readers. There are no comments on the content of the work. However, it seems to me that there are not enough references to the work of other researchers for the review. Separate sections of this review could be supported by additional references.

Author Response

The presented review characterizes the global world problems quite deeply. The material of the manuscript is built logically. I think the manuscript will be of interest to a wide range of readers. There are no comments on the content of the work. However, it seems to me that there are not enough references to the work of other researchers for the review. Separate sections of this review could be supported by additional references.

We have added in numerous details and references throughout the paper.

Reviewer 3 Report

The paper is clearly written and well organised. It provides a good overview of the research capacity that may be relevant for the transition towards a greener and more sustainable and circular economy in relation to the food system. The manuscript may be relevant to foster an open discussion to overcome barriers and fill the gaps that currently prevent a significant growth of the sector in LA.

Suggestions:

Although the authors cited a good number of references, I suggest to further improve the reference list for those readers who want to deepen the understanding of specific sectors and research areas. Below  a list of paragraphs that should br considered for additional citations: 

Lines 104-114

Lines 117-120

Line 125 (sentence endings with “plants and animals”)

Line 129 (sentence endings with “fertilisers”)

Line 147 ((sentence endings with “infrastructures”)

Lines 197-199

Line 260 (chemicals)

 Minor comments:                                                                                       

-        Lines 47-48. Please reformulate the sentence to improve clarity.

-        Modify table 1 in order to have all the text lines clearly visible.

-        The source (Chavarría et al. (2020)). should be moved to the table legend or reported as a note at the bottom of the table.

-        Lines 220-222. This sentence seems out of the context. I suggest to remove it from this point.

Author Response

The paper is clearly written and well organised. It provides a good overview of the research capacity that may be relevant for the transition towards a greener and more sustainable and circular economy in relation to the food system. The manuscript may be relevant to foster an open discussion to overcome barriers and fill the gaps that currently prevent a significant growth of the sector in LA.

Suggestions:

Although the authors cited a good number of references, I suggest to further improve the reference list for those readers who want to deepen the understanding of specific sectors and research areas. Below a list of paragraphs that should be considered for additional citations:

Lines 104-114 – references added

Lines 117-120 – reference added

Line 125 (sentence endings with “plants and animals”) – reference added

Line 129 (sentence endings with “fertilisers”) – references added

Line 147 ((sentence endings with “infrastructures”) – reference added

Lines 197-199 – reference added

Line 260 (chemicals)  reference added

 Minor comments:                                                                                      

-        Lines 47-48. Please reformulate the sentence to improve clarity.

This sentence has been revised.

-        Modify table 1 in order to have all the text lines clearly visible.

The table has been recreated.

-        The source (Chavarría et al. (2020)). should be moved to the table legend or reported as a note at the bottom of the table.

The source has been added to the table.

-        Lines 220-222. This sentence seems out of the context. I suggest to remove it from this point.

The sentence has been deleted.

Reviewer 4 Report

The manuscript presents a review of the possibilities of the bio-economy with respect to the increased sustainability of food systems. However, it is by no means suitable for publication. The major criticism lies in the fact that it reads more like a report or term paper than a scientific article.

There is no critical evaluation, no explicit methodology, and no new evidence. As the authors themselves state in the final paragraph, it is an "overview" (line 346). The structure of the paper is not suitable for a scientific paper, because it lacks all the elements to contribute to the scientific debate on the topic, as well is not structured in a logical and consequential manner. The conclusions (which are entitled 'Moving forward') are very sloppy and do not provide any elements for discussion.

Furthermore, the manuscript is graphically poor: the journal template was not used, the images are generalist (Fig. 1) and the tables were not formatted but simply copied/pasted from other sources (Table 1).

The manuscript, in this form, has little scientific value and it is therefore proposed to reject it and try another submission starting from a more solid base. 

Author Response

The manuscript presents a review of the possibilities of the bio-economy with respect to the increased sustainability of food systems. However, it is by no means suitable for publication. The major criticism lies in the fact that it reads more like a report or term paper than a scientific article.

There is no critical evaluation, no explicit methodology, and no new evidence. As the authors themselves state in the final paragraph, it is an "overview" (line 346). The structure of the paper is not suitable for a scientific paper, because it lacks all the elements to contribute to the scientific debate on the topic, as well is not structured in a logical and consequential manner. The conclusions (which are entitled 'Moving forward') are very sloppy and do not provide any elements for discussion.

Furthermore, the manuscript is graphically poor: the journal template was not used, the images are generalist (Fig. 1) and the tables were not formatted but simply copied/pasted from other sources (Table 1).

The manuscript, in this form, has little scientific value and it is therefore proposed to reject it and try another submission starting from a more solid base.

This article is a review article, which covers a large number of previously published articles, summarizing their findings and importance as they relate to the bioeconomy and food systems. By their very nature, there is never new evidence presented in a review article, nor is a methodology required, as no methodology is used to review the literature. The reviewer seems to not be familiar with review articles, their structure, their format and their contributions to the literature.